# A Combined Quantitative Evaluation Model for the Capability of Hyperspectral Imagery for Mineral Mapping

**DOI:** 10.3390/s19020328

**Published:** 2019-01-15

**Authors:** Na Li, Xinchen Huang, Huijie Zhao, Xianfei Qiu, Kewang Deng, Guorui Jia, Zhenhong Li, David Fairbairn, Xuemei Gong

**Affiliations:** 1School of Instrumentation Science and Opto-Electronic Engineering, Beihang University, Beijing 100191, China; gogo422@buaa.edu.cn (X.H.); qiuxianfei@buaa.edu.cn (X.Q.); dengkewang@buaa.edu.cn (K.D.); jiaguorui@buaa.edu.cn (G.J.); zhongkezhijie@163.com (X.G.); 2School of Engineering, Newcastle University, Newcastle NE1 7RU, UK; zhenhong.li@newcastle.ac.uk (Z.L.); david.fairbairn@newcastle.ac.uk (D.F.)

**Keywords:** hyperspectral imaging model, application capability, quantitative evaluation, normalized average distance

## Abstract

To analyze the influence factors of hyperspectral remote sensing data processing, and quantitatively evaluate the application capability of hyperspectral data, a combined evaluation model based on the physical process of imaging and statistical analysis was proposed. The normalized average distance between different classes of ground cover is selected as the evaluation index. The proposed model considers the influence factors of the full radiation transmission process and processing algorithms. First- and second-order statistical characteristics (mean and covariance) were applied to calculate the changes for the imaging process based on the radiation energy transfer. The statistical analysis was combined with the remote sensing process and the application performance, which consists of the imaging system parameters and imaging conditions, by building the imaging system and processing models. The season (solar zenith angle), sensor parameters (ground sampling distance, modulation transfer function, spectral resolution, spectral response function, and signal to noise ratio), and number of features were considered in order to analyze the influence factors of the application capability level. Simulated and real data collected by Hymap in the Dongtianshan area (Xinjiang Province, China), were used to estimate the proposed model’s performance in the application of mineral mapping. The predicted application capability of the proposed model is consistent with the theoretical analysis.

## 1. Introduction

Compared with multispectral or panchromatic images, the spectral resolution (SR) of hyperspectral images has been significantly improved. Information on both spectral and spatial dimensions can be acquired simultaneously [1]. As a result, hyperspectral techniques are widely used in many fields. However, the application performance of hyperspectral images is limited by several factors in the physical imaging process [2]. Predicting and quantitatively evaluating the application capability of hyperspectral data is necessary to improve its application performance, support improved sensor design, and provide satisfactory services to data users. And there are some other upcoming hyperspectral mission and the simulators. Before the satellite launching, it is necessary to simulate and evaluate the process by some end-to-end simulators like EnMap (Environmental Mapping and Analysis Program) [3], EeteS (EnMAP End-To-End Simulation Tool) [4] and FLEX (Fluorescence Explorer) [5].

In order to research the influence factors of imaging system parameters on application capability, a series of evaluation models are proposed. These methods include empirical analysis-, simulation model-, and analysis-based methods.

In the empirical analysis-based method, different parameter effects are analyzed from different images. Regression methods are used to analyze the relationships between system parameters and application capability. These include the general image quality equation [6], spectral quality equation [7], Kappa coefficient equation [8], probability of correct detection [9], and spatial−spectral total variation methods [10]. These methods can be used to evaluate data application capability, but they do not consider the imaging system principles, the relationships between system parameters, and the effects of processing algorithms.

In the simulation-based method, each pixel would be used as the research unit, and the pixel values can be obtained by simulation of the scene, the atmosphere, and the sensor. Here, simulated images are applied to classification or target detection, and then their application capability is evaluated [11,12]. These methods are applied to analyze the imaging performance of different scenes, but they are not combined with the application mission and do not consider the effects of processing algorithms.

The analysis-based model has become a research focus for the analysis of the influence factors of processing algorithms and of the full imaging process [13,14,15,16]. For this method, each class can be used as the research unit, and the statistical characteristics are introduced to describe the influence factors of imaging parameters and processing algorithms on the full imaging process. The index of application capability is derived from statistics [17]. The analysis model is applied to evaluate the capability of target detection or classification by calculating the effects of the scene, noise of the sensor, and processing [14]. The results are used to evaluate and predict the application performance of the airborne imaging spectrometer HYDICE (Hyperspectral Digital Imagery Collection Experiment) [14]. Parente and Clark used the analysis model to evaluate the accuracy of a hyperspectral imaging system in mapping the surface minerals of Mars [15]. The analysis model was extended to analyze and predict the application performance of an optical polarization imaging system [16]. However, these methods do not describe the quantitative relationship between the evaluation index of application capability and imaging parameters. Meanwhile, they do not consider the effects of spectral response functions (SRFs), such as spectral aliasing in the adjacent band.

Therefore, a combined quantitative evaluation model was proposed here. The application capability of hyperspectral data refers to the performance or the interpretation abilities of hyperspectral images for different applications. And the physical imaging process is combined with statistical analysis and the effects of application capability. They are analyzed together by building a scene (ground surface, atmosphere, and seasons), incorporating sensor models (spectral imaging, spatial imaging, and noise models), and applying processing algorithms (the number of features). The normalized average distance, which is a measure of class separability, is introduced to describe the quantitative relationship between the application capability and imaging parameters. The simulated and real data are used to evaluate the performance of the proposed model when applied to mineral mapping. We used mineral properties to explore the effects of different parameters on hyperspectral imagery’s capacity under macroscopic conditions. So that it’s not sensitive to BRDF (Bidirectional Reflectance Distribution Function) [18]. In the proposed model, application capability is a general definition, and can also be used in multi-spectral data and RGB images. The differences between them are the values which are calculated by different parameters, and the image impacts. The model provides guidance for the design and application of the next generation of hyperspectral remote sensing satellites.

GF-5 is a series of Chinese civilian remote sensing satellites meant to conduct scientific research on the earth’s atmosphere. The spectral ranges cover from VNIR (Visible and Near Infrared) to SWIR (Short-Wave Infrared) (400–2500 nm). The spatial resolution is 30 m. The spectral resolutions are 4 nm and 8 nm determined by spectral range. Real and simulated hyperspectral data with different parameters were used to estimate the capability of the proposed model in the Dongtianshan area (Xinjiang Province, China) [19]. The evaluation index of the application capability can be calculated with different imaging-processing parameters by using the proposed evaluation model. The imaging-processing parameters include SZA (Solar Zenith Angle), GSD (Ground Sampling Distance), the modulation transfer function, and the NSF (Number of Spaectral Features) of spectral features. The application capability can be estimated with different evaluation index values and compared with the system capability values calculated with the system-designed parameters, assuming the best imaging conditions.

## 2. imaging-processing Analytic Description

The physical imaging model is the basis for analyzing the application capability of the imaging parameters for hyperspectral data. According to the process of radiation transfer and the characteristics of spatial distribution of ground cover, the signals which are collected from ground targets, the background, and their statistical information will vary. Therefore, a full process model (including every class model, imaging conditions model, sensor model, and processing algorithm) is needed to calculate the statistical information of different influence factors. The evaluation index, created to determine application capability, is calculated by the statistical information (including mean and covariance) and then combined in a proposed quantitative evaluation model based on physical imaging and statistical information. The imaging–processing steps and analytic description are shown in Figure 1. The former involves the Earth’s surface class model, imaging condition model, sensor model, and processing algorithms. The analytic description is based on multivariate statistical analysis and the radiation transmission process. The mean and covariance are used to describe the changes of radiation in the imaging-processing process.

### 2.1. Class Model and Analytic Description of the Surface of the Earth

Reflectance is one of the inherent characteristics of each class of ground cover. According to the analysis of surface characteristics and the distribution of samples, the normal distribution model of each ground class is considered in our work [18]. Thus, first- and second-order statistical characteristics can be used to describe all statistical characteristics of each class in the image classification. Therefore, mean and covariance were used to build the surface class model of Earth. For class *k*, the mean reflectance value $\bar{X_{k}}$ and covariance $\Sigma_{k}$ can be calculated with its reflectance of the training samples based on the ground class model.

### 2.2. Model of Imaging Conditions and the Analytic Description

The imaging conditions identified for this project include imaging date, location, season, and atmospheric properties. These will affect the at-sensor radiance and subsequently influence the calculation of application capability. The effects of these factors can be described by the atmospheric radiation transfer model.

The radiation actions of the atmosphere strongly influence reflection, absorption, and scattering [20]. Under these influences, the at-sensor radiance can be divided into three parts; namely, direct reflection of the target, path radiation, and the radiation reflected from the nearby background [21]. The details of the atmosphere transfer model adopted here are shown as follows:(1)$$\begin{array}{l} L_{\lambda }=L_{\lambda ,s}\cdot X+L_{\lambda ,path}^{}+\left(L_{\lambda ,1}^{}-L_{\lambda ,path}^{}\right)\cdot X_{e} \end{array}$$
(2)$$L_{\lambda ,s}=\frac{1}{\pi }\cdot \frac{\cos\left(\theta_{solar}\right)E_{\lambda ,Direct}}{\exp\left(-1.26\tau_{\lambda }\right)}\cdot T_{\lambda ,atm}$$
(3)$$\tau_{\lambda }=1.35\sec\left(\theta_{solar}\right)\lambda^{-1.328}V_{\eta }{}^{-0.656}$$
where *L_λ_* is the at-sensor spectral radiance, *L_λ,s_* is the radiation reaching the surface of the Earth, X is the reflectance vector of objects, *L_λ_*_,*path*_ is the path radiation, *L_λ,_*_1_ is the radiation that removes the direct reflection of objects when the surface reflectance is 1, *X_e_* is the reflectance vector of the background, *τ_λ_* is the atmospheric optical thickness and is calculated by imaging location, imaging time, and atmospheric conditions, *E_λ,Direct_* is the solar spectral direct irradiance vector and is a constant calculated by atmospheric radiation transmission model (such as MODTRAN, Moderate resolution atmospheric transmission), *T_λ,atm_* is the atmospheric spectral transmittance vector, *θ_solar_* is the solar zenith angle (SZA), and *V_η_* is the atmospheric visibility.

Therefore, the mean $\bar{L_{\lambda }}$ and covariance $\Sigma_{L_{\lambda }}$ of the at-sensor radiance can be calculated based on the following formulas (4) and (5):(4)$$\bar{L_{\lambda }}=L_{\lambda ,s}\cdot \bar{X}+L_{\lambda ,path}^{}+\left(L_{\lambda ,1}^{}-L_{\lambda ,path}^{}\right)\cdot \bar{X_{e}}$$
(5)$$\sigma_{L,mn}=L_{m,s}\sigma_{X,mn}L_{n,s}+\left(L_{m,1}^{}-L_{m,path}^{}\right)\sigma_{e,mn}\left(L_{n,1}^{}-L_{n,path}^{}\right)$$
where $\bar{L_{\lambda }}$ is the mean of *L_λ_*; $\bar{X}$ is the reflectance mean of one class; $\bar{X_{e}}$ is the mean of *X_e_*; *σ_L,mn_* is the value of the mth row and nth column of covariance of *L_λ_*; *L_m,s_* and *L_n,s_* are the m- and n-band values of *L_λ.s_*, respectively; and *σ_X,mn_* and *σ_e,mn_* are the values of the mth row and nth column of the reflectance covariance matrix $\Sigma_{L_{\lambda }}$ of one of classes, and the mean reflectance covariance matrix of all backgrounds. Therefore, the SZA is used to analyze the effect of the imaging condition.

### 2.3. Sensor Model and Analytic Description

The incident radiation will be separated in parallel through the front optical system. First, it is collected by the detector, after which it is amplified and quantified by the electronic system. Therefore, the sensor contains all the spatial, spectral, and radiometric effects present in converting at-sensor spectral radiance into a digital number of hyperspectral imaging data. A spatial model, a spectral model, and a noise model of the sensor for the hyperspectral imaging system are required to quantitatively analyze the effects of the sensor and implement an analytic description of the process.

#### 2.3.1. Spatial Model and Analytic Description

The influences of the spatial model of the sensor on hyperspectral data mainly include the reduction of energy entering the sensor owing to the effects of the transmittance of the optical system and the quantum efficiency, and image degradation because of the influence of the point spread function [22]. Meanwhile, the Gaussian model is introduced to describe the distribution of the point spread function according to the characteristics of the imaging system. The total energy collected by the sensor and the total number of pixels of the imaging system remained unchanged. Therefore, the mean of the radiation collected after spatial model $\bar{L_{L_{\lambda }}^{s}}$ will be affected by the coefficients of radiation reduction, and will not have any effect on image degradation. The covariance $\overset{s}{\underset{L_{\lambda }}{\sum }}$ will be affected by both the coefficients of radiation reduction and the form of the point spread function [23]. The analytic description is shown as follows: (6)$$\bar{L_{L_{\lambda }}^{s}}=A\cdot \bar{L_{\lambda }}$$
(7)$$\Sigma_{L_{\lambda }}^{s}={\left[\sigma_{L,mn}^{s}\right]}_{N\times N}={\left[W_{s}^{mn}\sigma_{A,mn}\right]}_{N\times N}$$
(8)$$\begin{array}{c} \Sigma_{A}=A\Sigma_{L_{\lambda }}A^{T}\\ W_{s}^{mn}=4\exp\left[\left(a_{mn}^{2}+b_{mn}^{2}\right)\sigma_{1}\sigma_{2}\right]\cdot gf\left(a_{mn}\cdot \sqrt{2}\sigma_{1}\right)\cdot gf\left(b_{mn}\cdot \sqrt{2}\sigma_{2}\right)\\ gf(\alpha )=\frac{1}{\sqrt{2\pi }}\int_{\alpha }^{+\infty }e^{-\tau^{2}/2}d\tau \end{array}$$
where *A* is the coefficient vector of radiation reduction after the spatial model, *N* is the number of bands, *σ_A,mn_* is the value of the mth row and nth column of the covariance matrix $\Sigma_{A}$, $W_{s}^{mn}$ is the value of the mth row and nth column of weight matrix *W_s_*, and $\sigma_{1}$ and $\sigma_{2}$ are the standard deviations of the point spread function in the along- and across-orbit directions, respectively.

Formulas (6) to (8) show the effects of the spatial model that can be described by radiation reduction and image degradation with the Ground Sampling Distance (GSD) and point spread function. The Modulation Transfer Function (MTF) could be calculated by the Fourier transform of point spread function. Therefore, the GSD and MTF were introduced to analyze the effects of the spatial model for application capability, and $\sigma_{1}$ and $\sigma_{2}$ can be estimated and measured from the image. Then, the relationship between the imaging system parameters and application performance was established.

#### 2.3.2. Spectral Model and Analytic Description

The radiation received by each band of the sensor includes not only the center-wavelength radiation, but also the adjacent wavelengths radiation. Therefore, a certain percentage of the radiation will be allocated to several adjacent bands, and the values detected in each band are also the result of the contribution of adjacent bands [24]. This process can be described by SRF. Here, the spectral model applied has the same SR as the Earth surface model for the imaging spectrometer [25]. The spectral effects of a sensor can be described analytically by a linear transformation matrix *B*, which consists of the SRF. Therefore, the mean $\bar{S}$ and covariance $\Sigma_{s}$ of the sensor-received signal vector are described as follows:(9)$$\bar{S}=B\cdot \bar{L_{L_{\lambda }}^{s}}=B\cdot A\cdot \bar{L_{\lambda }}$$
(10)$$\Sigma_{s}=B\cdot \Sigma_{L_{\lambda }}^{s}\cdot B^{T}=BW_{S}^{}A\Sigma_{L_{\lambda }}A^{T}B^{T}$$
(11)$$B=\Delta \lambda {\left[\begin{array}{l} 1\_res\\ 2\_res\\ \vdots \\ N\_res \end{array}\right]}_{N\times N}$$
where Δλ is the SR of the spectral radiance vectors, and *N_res* is the value of SRF in the Nth band.

Formulas (9) to (11) show that the effects of the spectral model can be described by the SR and SRF of the imaging system, and these two parameters were selected to analyze the effects for application capacity and to build the relationship between the parameters of the imaging system and the application performance.

#### 2.3.3. Noise Model and Analytic Description

The noise model is built from an integration of random and system noise processes. The random noises are zero-mean noises and include the photon noise, read-out noise, and quantization error [26]. The system noises are non-zero mean noises, which include dark-current noise and radiometric error [27]. Therefore, the effects of dark-current noise $n_{dark}$ and radiometric error $e_{rad}$ can be added directly to the noisy mean vector $\bar{Y}$ as follows:(12)$$\bar{Y}=\left(1+e_{rad}\right)\bar{S}+n_{dark}=\left(1+e_{rad}\right)B\cdot A\cdot \bar{L_{\lambda }}+n_{dark}$$

The covariance of random noises can be calculated by using their statistical distribution characteristics, and they can be added directly. $\Sigma_{Y}$ is the signal covariance vector after noise model and it can be calculated as follows:(13)$$\begin{array}{l} \Sigma_{Y}={\left(1+e_{rad}\right)}^{2}\Sigma_{s}+\Lambda_{dark}+\Lambda_{pho}+\Lambda_{read}+\Lambda_{quant}\\ ={\left(1+e_{rad}\right)}^{2}B\cdot W_{S}^{}\cdot A\cdot \Sigma_{L_{\lambda }}\cdot A^{T}\cdot B^{T}+\Lambda_{dark}+\Lambda_{rand} \end{array}$$
where, $\Lambda_{pho}$, $\Lambda_{read}$, and $\Lambda_{quant}$ are the covariance matrices of photon noise, read-out noise, and quantization error, respectively, and $\Lambda_{rand}=\Lambda_{pho}+\Lambda_{read}+\Lambda_{quant}$ is the covariance matrix.

Formulas (12) and (13) show that the noise level of the imaging system can describe the effects of the noise model. In our work, the signal-to-noise ratio (SNR) was applied to analyze the effects of noise level for the evaluation of the application performance.

### 2.4. Model of data processing and Analytic Description

In the data processing, the main methods include the feature extraction and selection methods and the detection or classification methods. The detection or classification methods cannot change the statistical characteristics, therefore, the feature extraction and selection methods will affect the data application performance. Therefore, these effects will be modeled in this section. Feature extraction and selection will reduce the bands of data, and will also change the mean and covariance of the signal vector. A weighting or transformation matrix *F* can be considered to react to the signal vector [28]. Therefore, the number of spectral features (NSF) was used to describe the influence factors of the processing algorithm for the application capability. The mean $\bar{Z}$ and covariance $\Sigma_{z}$ after processing, can be calculated as follows:(14)$$\bar{Z}=F\bar{\cdot Y}=F\left(1+e_{rad}\right)B\cdot A\cdot \bar{L_{\lambda }}+F\cdot n_{dark}$$
(15)$$\begin{array}{l} \Sigma_{z}=F\cdot \Sigma_{Y}\cdot F^{T}\\ =F\cdot \left[{\left(1+e_{rad}\right)}^{2}B\cdot W_{S}^{}\cdot A\cdot \Sigma_{L_{\lambda }}\cdot A^{T}\cdot B^{T}+\Lambda_{rand}\right]\cdot F^{T} \end{array}$$

In Figure 1 is the flowchart for whole imaging-processing progress and the analytic description. It mainly considers the energy transfer throughout the model. The imaging condition model affects the at-sensor radiance and influences the application capability. The loading system resulted in a digital number. We can obtain the spectral signal statistic from the sensor. Thus, with the signal changing the first- and second- order statistics (mean and covariance) can be used to describe all the statistical characteristics of each class in the image classification.

## 3. Combined Quantitative Evaluation Model

### 3.1. Evaluation Index

Figure 2 shows the classification error varies with the normalized distance. x represents the reflectance of ground object in one wavelength. And P represents the probability corresponding to this value. Two Gaussian curve represents two different classes’ spectra. The overlap represents the error probability. In this figure, we can see the inter-class variance of the two samples remained the same and the between-class distance increased, the probability of classification errors decreases; However, when the mean distance between the two classes is same and the intra-class variance increases, the error probability of classification increases.

The separability of different ground cover classes is important when the hyperspectral data are used for different applications (such as target detection, classification, and identification). According to the Bayes criteria, if within-class dispersion is relatively smaller, and inter-class dispersion is larger, the separability of different ground cover classes is improved and the probability of error in object detection or ground cover classification is minimized [29]. Therefore, the normalized mean distance was applied as the evaluation index for hyperspectral remote sensing data applications, to obtain the general evaluation parameters in different applications.

The evaluation index *D* is shown in Equation (16), as follows:(16)$$D=\sqrt[{\frac{m\left(m-1\right)}{2}}]{\prod_{i=1}^{m}\prod_{j=1>i}^{m}D_{ij}}$$
(17)$$D_{ij}=\sqrt{\sum_{k=1}^{N}d_{k}{}^{2}},\;d_{k}=\frac{\left|\mu_{1,k}-\mu_{2,k}\right|}{\sqrt{\Sigma_{1,kk}}+\sqrt{\Sigma_{2,kk}}}$$
where *D_ij_* is the normalized mean distance between the ith and jth classes, *m* is the number of classes, *d_k_* is the normalized mean distance in the kth band with two classes, $\mu_{1,k}$ and $\mu_{2,k}$ are the means of different kinds of minerals in the kth band, and $\Sigma_{1,kk}$ and $\Sigma_{2,kk}$ are the covariance of each mineral in the kth band.

### 3.2. Proposed Joint Quantitative Evaluation Model

The combined evaluation model, based on statistical analysis and the process of physical imaging, is proposed for hyperspectral images delivered by the Chinese GF-5 satellite. In our model, the mean and covariance of each class are derived and calculated. The mean and covariance of the signal vector are collected from the sensor by analyzing the influence factors of the imaging system parameters and then calculating the mean and covariance after feature selection and extraction. Therefore, the proposed evaluation model can quantitatively describe the influence factors of the physical imaging process and processing algorithm. It can also implement integration of statistical analysis and the imaging-processing process. By applying the proposed evaluation index and imaging-processing analytic description, the combined evaluation model can be presented as follows:(18)$$D=\sqrt[{\frac{m\left(m-1\right)}{2}}]{\prod_{i=1}^{m}\prod_{j=1>i}^{m}\sqrt{\frac{1}{q}\sum_{k=1}^{q}(\frac{\left|\mu_{i,k}-\mu_{j,k}\right|}{\sqrt{\Sigma_{i,kk}}+\sqrt{\Sigma_{j.kk}}}}{)}^{2}}$$
where *q* was the NSF obtained by the feature extraction or selection algorithm based on the processing model, and the mean $\mu_{i}$ and covariance $\Sigma_{i}$ could be calculated as follows:(19)$$\begin{array}{l} \mu_{i}=F\left(1+e_{rad}\right)B\cdot A\cdot \left[\begin{array}{l} \frac{1}{\pi }\frac{\cos\left(\theta_{solar}\right)E_{\lambda ,Direct}}{\exp\left(-1.26\tau_{\lambda }\right)}\cdot T_{\lambda ,atm}\cdot \bar{X_{i}}\\ +L_{\lambda ,path}^{}+\left(L_{\lambda ,1}^{}-L_{\lambda ,path}^{}\right)\cdot \bar{X_{e}} \end{array}\right]+F\cdot n_{dark}\\ \Sigma_{i}=F\cdot \left[{\left(1+e_{rad}\right)}^{2}B\cdot W_{i,S}^{}\cdot A\cdot \Sigma_{i,L_{\lambda }}\cdot A^{T}\cdot B^{T}+\Lambda_{dark}+\Lambda_{rand}\right]\cdot F^{T} \end{array}$$

## 4. Experiment and Results Analysis

### 4.1. Experiment Data

The airborne hyperspectral data collected by Hymap are applied to the mining area in Dongtianshan (Xinjiang Province, China) to evaluate the validity and applicability of the proposed model. Approximately nine types of altered minerals are found in the research area (with some types existing as mixed pixels): calcite, chlorite, aluminum-rich sericite, montmorillonite, taxoite, salinization, talc, and their mixed minerals.

In order to analyze the different influence factors of application capability, which are caused by the imaging system parameters, a total of 50 images were simulated using a satellite-borne hyperspectral image simulation method. A Hymap reflectance image was used as the input scene. The several processing steps were used to generate the simulated images, including the creation of atmospheric effects using an improved radiation transfer model that considers topographic effects [17], the satellite-borne imaging spectrometric model which incorporates the spectral, radiative, and spatial features of the hyperspectral imager [9], and the simulation of a set of data with different imaging system parameters (SNR, SR, GSD, SRF, and MTF) under the same atmospheric and imaging conditions and with different SZA under the same imaging system parameters were simulated in the simulation experiment. And then the mean and covariance with different parameters were calculated to obtain the different the evaluation index of application capability based on our proposed combined evaluation model. According to geological data and prior research results, the samples of different minerals were selected (Figure 3) and applied to calculate the mean and covariance of different ground cover classes. The imaging condition parameters of the simulated data are shown in Table 1.

### 4.2. Analysis of Results

An evaluation index was initially calculated based on the highest SZA (66.56°) and the designed system parameters (shown in Table 2), to compare and analyze the application capability. Its value in these conditions is 20.65 (shown in Figure 3). This evaluation index was calculated at regular intervals with tuning of only one of the imaging conditions and system parameters.

#### 4.2.1. Imaging Condition-SZA

SZA can be calculated by imaging date, imaging time, and research area location. Therefore, the effects of imaging conditions (such as seasons and location) can be considered by using the factor of SZA. Considering the research area location, the minimum and maximum values of SZA are 21° and 66.56°, respectively. The values of the combined evaluation model were calculated with different SZAs (set at 25°, 35°, 45°, 50°, 60°, and 65°) and the parameters from Table 2. The results in Figure 4 show that the evaluation index decreased from 22.7 to 20.73 with increasing SZA from 25° to 65°. The main reasons for this include the fact that the solar radiation energy collected by the sensor weakens, and the detailed spectral features and their differences cannot be identified with increasing SZA.

#### 4.2.2. Hyperspectral Imaging Parameters

The main hyperspectral imaging parameters selected to analyze the influence factors for the application capability include GSD, MTF, SR, SRF, and SNR. The detailed analysis of imaging parameters, carried out in order to analyze application requirements and payload manufacture performance, is shown in Figure 4 and as follows.

GSD is an important parameter for distinguishing the spatial distribution of different classes. For the mineral mapping application, GSD was set in this project from 10 to 60 m. Therefore, the combined evaluation model was calculated with different GSDs (10, 20, 30, 40, 45, and 60 m) under the condition of the minimum SZA and above-mentioned parameters (shown in Table 2). Figure 4a,b show the result of the mineral mapping application, and demonstrates that the evaluation index increases initially and then decreases with the rise of GSD (the evaluation index is maximized when the value of GSD is 40 m). The main reason for this is the higher GSD and the lower energy in one unit collected by the imaging system. Meanwhile, when the value of GSD increases, the number of mixed pixels increases. Therefore, the above-mentioned two reasons result in decreased application capability, consistent with the calculated results shown in Figure 4a,b.

The MTF is the magnitude of Fourier transform of the point spread function, and can be used to describe the definition of a hyperspectral image in spatial dimension. Most specifications are written in terms of MTF as a function of spatial frequency. The MTF will decrease with the increase of the standard deviation of the point spread function. Considering the capability of payload manufacture, an MTF greater than 0.45 at Nyquist frequency is technically difficult to achieve. Nyquist frequency depends on GSD—it equals to 1/(2*GSD). The MTF at Nyquist is a measure of aliasing. From the point of view of user requirements, the quality of the hyperspectral image is very unacceptable and cannot be used when the MTF is less than 0.05. Therefore, the MTF at Nyquist frequency ranges from 0.05 to 0.45 in our work, in order to calculate the joint evaluation model. The result was achieved in the conditions of the above-mentioned system parameters (shown in Table 2) and the lowest SZA with different MTF values, as shown in Figure 4a,c. The result shows that the evaluation index increases linearly with the increase of MTF. When the MTF at the Nyquist frequency is small, the standard deviation of the point spread function is large. That is, the collected data of each pixel is influenced by the surrounding pixels. Therefore, the spectrum of each class is affected by the surrounding pixels strongly, and the data application capability decreases. When the MTF at Nyquist frequency increases, the standard deviation of the point spread function decreases. The spectral features are weakened by the influence of the surrounding pixels, which are easier to identify, thereby improving application capability. When the MTF is greater than 0.16, the evaluation index is better than 20.65.

The SR is an important parameter to identify different classes directly through spectral features. The spectral feature width of minerals is from 20–40 nm. Therefore, the SR should be better than 10–20 nm. In our work, the value of different SRs ranged from 10–30 nm with a 5 nm step. The joint evaluation model was calculated under the conditions of the lowest SZA and the above-mentioned system parameters, in order to analyze the influence factors of application capability with different SRs (shown in Table 2). The calculated result in Figure 4a,d shows that the relationship between SR and the evaluation index is similar to the relationship between the GSD and the evaluation index. When the SR is lower, the detailed differences of the minerals cannot be identified, and the evaluation capability decreases. When the SR is too high, the energy collected by the imaging unit is very small, and the detailed spectral features are submerged by noise. Therefore, the evaluation index increases with the rise of the SR value when the SR is better than 15 nm. On the contrary, the evaluation index decreases with the rise of the SR value when the SR is lower than 15 nm.

The SRF should ideally be a rectangular function, without changing the radiation distribution in the spectral dimension collected by the detector. However, the SRF is mostly a Gaussian function in the actual imaging process, and the spectral dimension data of each band is affected by the adjacent bands. When the standard deviation of the SRF is better than 0.5, it is close to the ideal state, and the spectral energy distribution of each band is not affected by the adjacent bands. When the standard deviation of the SRF is greater than 2.5, the spectral aliasing is serious, and the data are difficult to apply. Therefore, the standard deviation of the SRF was from 0.5–2.5 with a 0.5 step in our research, in order to analyze the effects of the application capability with different SRFs. The joint evaluation model was calculated under the conditions of the lowest SZA and the above-mentioned system parameters, as shown in Table 2. The calculated result is shown in Figure 4a,e. When the standard deviation of the SRF is small, the spectral data of each band is weakly affected by the adjacent bands, and the detailed features can be detected, which is beneficial to identifying different minerals. Therefore, the application capability is higher. However, when the standard deviation of the SRF increases, the spectral data of each band is strongly affected by the adjacent bands—that is, the spectral curve is processed by a low-pass filter. Therefore, the detailed features cannot be obtained, and the difficulty of mineral identification and the application capability both decrease. The application capability will decrease if the standard deviation of the SRF increases. Therefore, the evaluation index is better than 20.65 when the standard deviation is better than 1.5, and the evaluation index increases with an increasing standard deviation value. The decrease of the evaluation index with a standard deviation from 0.5–1.5 is greater than that from 1.5–2.5.

The SNR is an important parameter for obtaining detailed spectral and spatial information, and is a very sensitive parameter for application capability. Considering the application requirement of mineral identification, the SNR should be better than 100@VNIR and 50@SWIR. Considering the capability of payload manufacture, obtaining 300@VNIR and 250@SWIR it is too difficult for a satellite-based hyperspectral imaging system. Therefore, the SNR was from 100@VNIR and 50@SWIR to 300@VNIR and 250@SWIR, with a 50 step, in order to analyze the effects of application capability with different SRFs. The joint evaluation model was calculated under the conditions of the lowest SZA and the system parameters shown in Table 2. The evaluation index with different SNRs was calculated, and the results were shown in Figure 5a,f. 

When the SNR is low, the noise level of the imaging system is high, and the detailed features are submerged in the noise and cannot be extracted. Identifying different minerals with similar spectral features is difficult, and the application capability of the data will then decrease. As the SNR increases, and if the signal does not change, the noise level decreases and the fine spectral features can be detected, thereby improving the data application capability. Therefore, the evaluation index increases with increasing SNR, and the value of the evaluation index is better than 20.65 when the SNR is greater than 150@VNIR and100@SWIR.

#### 4.2.3. Data Processing-NSF

A high correlation was observed among the adjacent bands in the hyperspectral narrow-band spectral data. In the case of the number of training samples being determined, the application capability will decrease as the bands of hyperspectral data increase. Therefore, hyperspectral data will be pre-processed with the dimension reduction model. However, the NSF must be calculated with different feature selection and extraction methods. The NSF values can be detected after feature extracting. The evaluation index was calculated with varying numbers of spectral features to analyze the effects of the NSF on the application capability. As the principal component analysis method was used to extract spectral features in our work, the minimum NSF was set to 5, and the amount of spectral information did not increase when the NSF was greater than 25. Therefore, the NSF ranged from 5 to 25 with 5 steps (Figure 6). With increased information on the original data that can be retained after the feature extraction with increasing NSF, the value of the data application capability increases. However, a Hughes phenomenon was observed in the data classification, as follows: increasing the NSF beyond a certain number leads to a decrease in the application capability [30]. In practice, the results show that the evaluation index increases with NSF being considered (5 to 20), and the evaluation index decreases when the NSF is greater than 20.

### 4.3. Discussion

The application capability is affected by imaging conditions, sensor parameters, and data processing methods for hyperspectral imaging system. The relationship among them can be obtained from the above-mentioned analysis, as shown in Table 3. The relationship between the application capability and SZA, MTF, SNR, and SRF (σ) is linear. Meanwhile, a monotonically increasing relationship is observed between the application capability and MTF and SNR. A monotonically decreasing relationship is observed between the application capability and SZA and SRF (σ). The relationship between the application capability and SR, GSD, and NSF is nonlinear. It is an approximate exponential relationship.

The sensitivity analysis between the evaluation index and parameters of the whole process (including imaging conditions, sensor parameters, and data processing) is shown in Figure 6. The SNR has the largest sensitivity and the largest effect among the seven parameters used for the evaluation index, followed by the NSF, GSD, and SR. The SNR is clearly very important in the hyperspectral imaging system design, and other parameters can be traded off against each other based on application requirements and payload manufacture. The correlation of different parameters should be considered in the design of the imaging system, satellite orbits, and the data processing system. The relationship can be applied to optimally design parameters for the imaging system, and in predicting the application performance.

Figure 7 is a boxplot which describes the whole process parameters clearly. It shows the data dispersion of every parameter that has been mentioned. The edges of the box are the 25th and 75th percentiles; the black lines on the top and bottom are the maximum and minimum values; and the red line is the median.

## 5. Conclusions 

In this work, a combined quantitative evaluation model based on statistical analysis and the physical imaging process is proposed. The normalized mean distance is used as the evaluation index, and the influence factors of seven parameters (SZA, MTF, SNR, SR, SRF, GSD, and NSF) are analyzed based on radiation transfer in the physical imaging process and data processing. The designed parameters of the GF-5 satellite and simulated hyperspectral data are used to estimate the proposed model performance. Furthermore, the relationship between the application capability, imaging conditions, imaging parameters, and processing algorithms is examined. A linear relationship exists between the application capacity and SZA, MTF, SNR, SRF, and an approximate index relationship exists between the application capacity and SR, GSD, and NSF. In the future, the joint effects and optimization of multi-parameters for the application capability will be studied. A potential application of this study is trade off design. Taking into account these seven parameters, the joint effects and optimization will promote application capability, potentially reducing costs. The internal parameters of the imaging spectrometer can be adjusted quantitatively to optimize the imaging effect. Another potential use of our research is in the pre-evaluation of the application before it is put to use. With the development of our research, several new kinds of parameters will emerge in the future, in order to better consummate the application capability. Additionally, we will take non-ideal imaging conditions into consideration, so that the research will be more comprehensive.

## Figures and Tables

**Figure 1 sensors-19-00328-f001:**
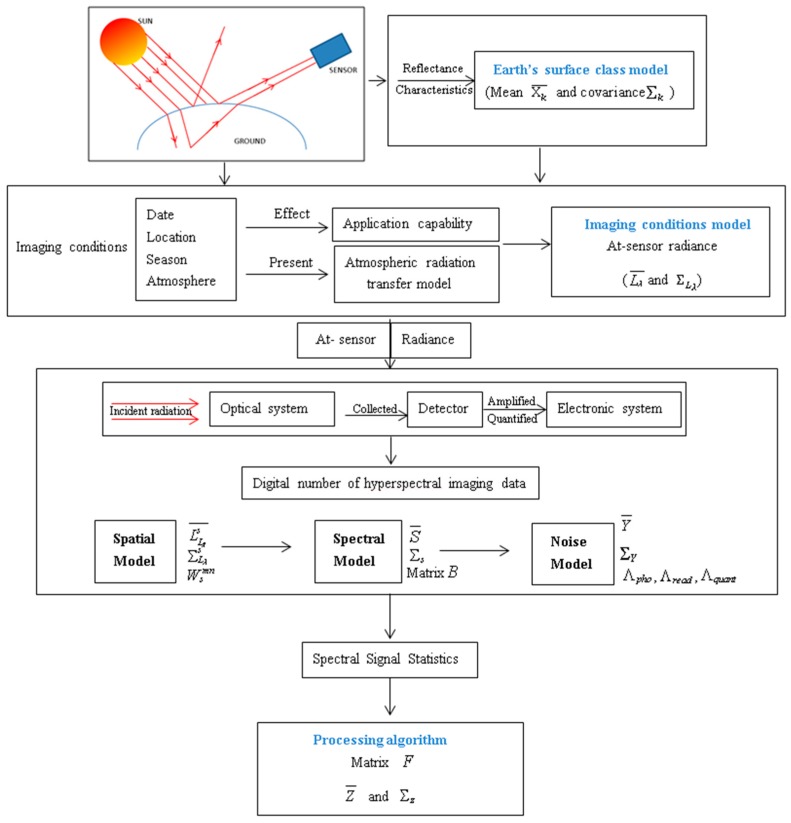
Flowchart for imaging-processing progress and the analytic description.

**Figure 2 sensors-19-00328-f002:**
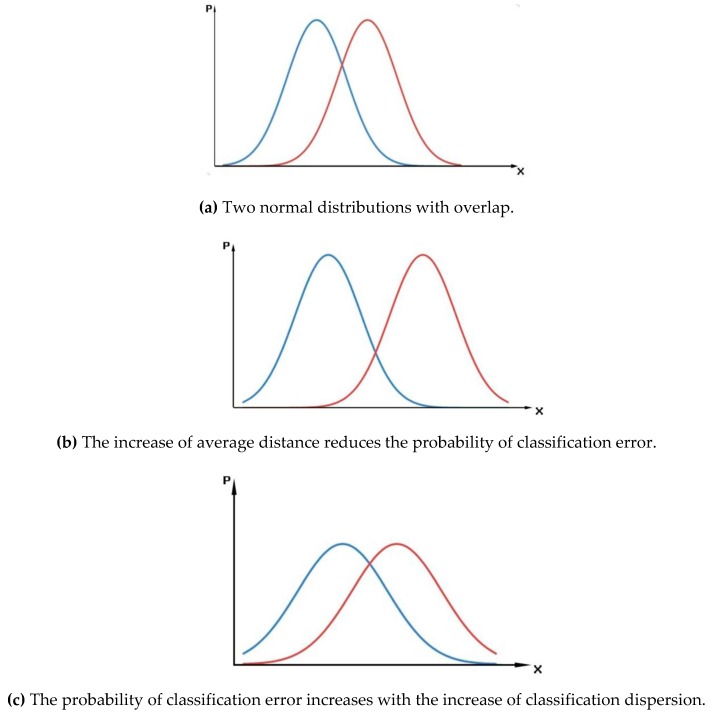
Classification error varies with the normalized distance.

**Figure 3 sensors-19-00328-f003:**
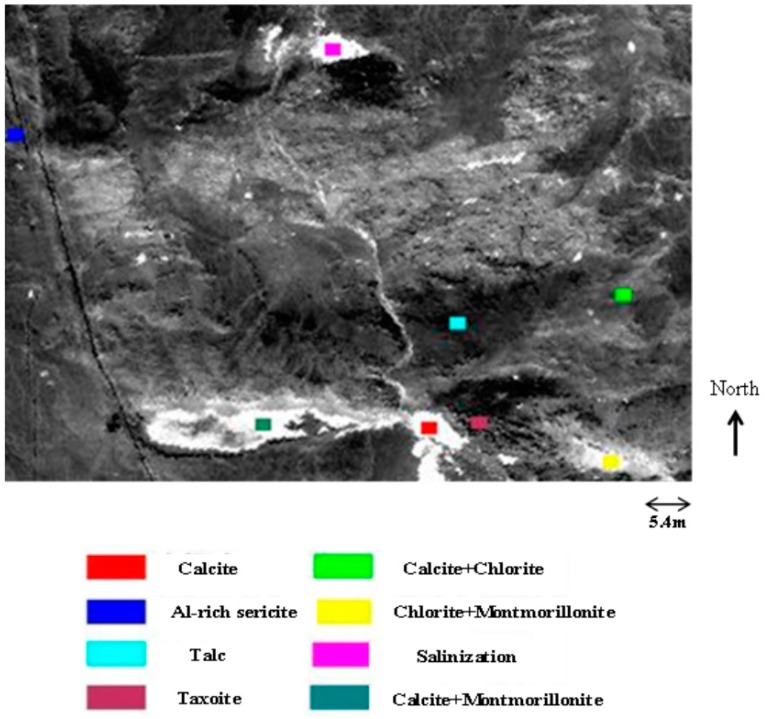
Different minerals made up training samples are taken into calculation.

**Figure 4 sensors-19-00328-f004:**
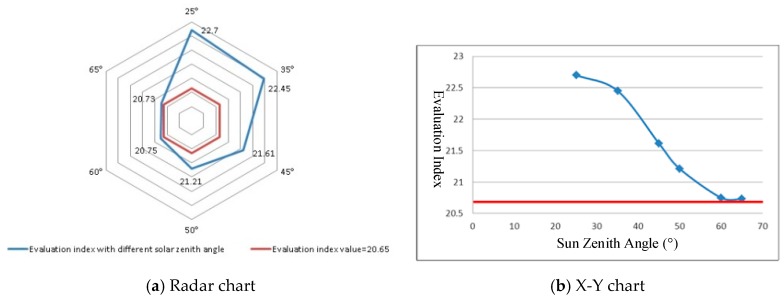
Evaluation index with different SZAs (Red line: 20.65).

**Figure 5 sensors-19-00328-f005:**
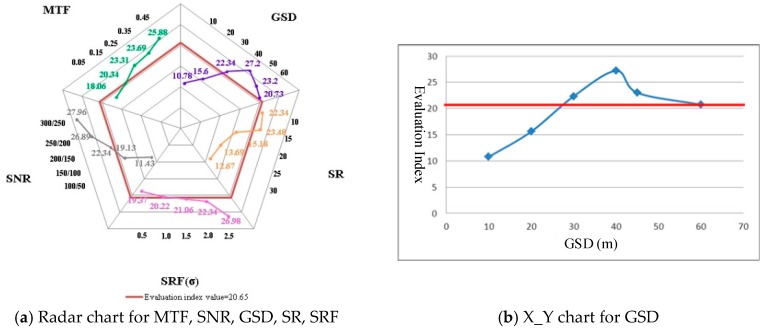
The evaluation index with different imaging parameters (Red line: 20.65).

**Figure 6 sensors-19-00328-f006:**
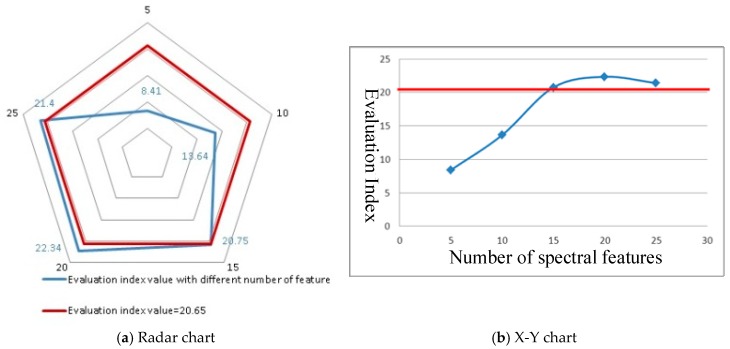
Evaluation index with different NSFs (Red line: 20.65).

**Figure 7 sensors-19-00328-f007:**
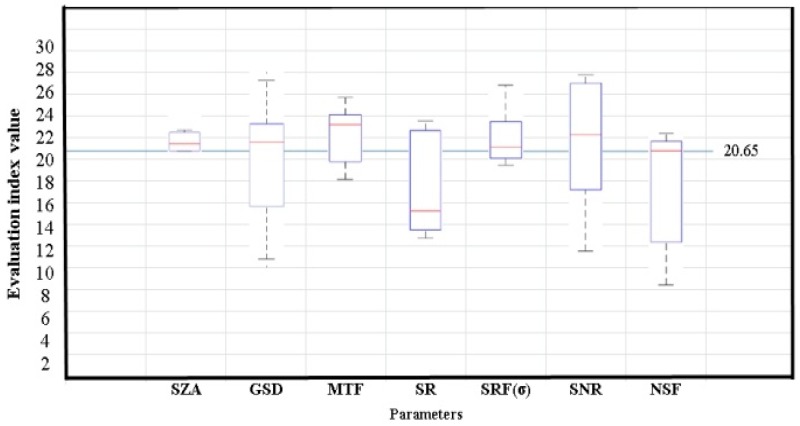
Sensitivity analysis between the evaluation index and the whole process parameters.

**Table 1 sensors-19-00328-t001:** Imaging conditions of the simulated data.

**Atmospheric conditions**	**Parameter values**
Atmospheric model	1976 US standard
	Mid-latitude Summer
Aerosol model	Rural
Visibility (km)	23
Adjacency effect radius (km)	1
Number of DISORT streams	8
**Imaging conditions**	**Parameter values**
A Height of orbit (km)	705
B Imaging date	1 June 2010
C Imaging time	GMT 6:30
D Ground elevation (km)	2
E View zenith angle	180°
F View azimuth angle	0°
G Location of imaging area	42.444° N 93.907° E, 42.372° N 94.296° E 42.029° N 93.768° E, 41.957° N 94.154° E

**Table 2 sensors-19-00328-t002:** Imaging conditions and designed system parameters.

Parameters	Value
GSD (m)	30
SR (nm)	10
SNR	200@VNIR/150@SWIR
SRF (σ)	1.0
MTF@Nyquist frequency	0.3
NSF	20

**Table 3 sensors-19-00328-t003:** Relationship between the application capability and parameters for mineral mapping.

Parameters (Increase)		Application Capability	Evaluation Index ≥20.65
Imaging conditions	SZA	↓	≤66.56°
Sensor	MTF	↑	≥0.15
	SNR	↑	≥150@VNIR and 100@SWIR
	SRF (σ)	↓	≤1.5
	SR (~15 nm)	↑	[10 nm 17.5 nm]
	SR (15 nm~)	↓	
	GSD (~40 m)	↑	[27.5 m 60 m]
	GSD (40 m~)	↓	
Processing	NSF (~20)	↑	[15 25]
	NSF (20~)	↓	

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
