# Peer review of "A Combined Quantitative Evaluation Model for the Capability of Hyperspectral Imagery for Mineral Mapping"

_sensors, 2019, doi:10.3390/s19020328_

Round 1

Reviewer 1 Report

Notwithstanding I have published two journal articles on an end-to-end hyperspectral simulator, I found the manuscript very hard to read and follow. The idea of extending the simulator to include also a supervised thematic classification is interesting, but introduces an further complexity. Although the dependence of the outcome of classification on the system and acquisition parameters is mostly known, the added value of the extended system may be of interest, though for few readers. 

For a deeply revised version of the manuscript, I recommend:

1) insert a table of symbols 

2) insert a table of acronyms

3) use suitable typesettings to indicate scalars, vectors, matrices, dependence on space  and on wavelenght; in the present manuscript, it is not allowed to know is the wavelegth lambda is a continuous or a discrete variable. I checked the notation of some articles that have been cited and it is much better.

4) Fig. 1 should possibly be moved at the end of Sect. 2 (not mandatory if the symbol table is clear and complete).

5) use compact symbol names, otherwise a formula will be unreadable. I wish to remark that most of equations are hardly readable also for an expert, due to poor notation.

6) The reference list should be extended to comprise other ucoming hyperspectral missions (not only EnMap) and possibly the simulators of their instruments.

Author Response

Manuscript ID: sensors-411757

Dear Professor,

Many thanks for the constructive and encouraging comments on our manuscript from three reviewers. In particular, we welcome and note your comment The idea of extending the simulator to include also a supervised thematic classification is interesting, but introduces a further complexity.” We enclose a carefully revised manuscript according to the comments and suggestions made. We provide an item-by-item response to all comments. The responses are start with “Response” which is included in “bold”. We have also made some non-requested minor typographical and readability edits.

We hope that these clarifications and revisions will now enable the paper to be accepted for publication in Sensors, and look forward to hearing from you soon.

Yours sincerely,

Na Li (on behalf of all authors)

Response :

Notwithstanding I have published two journal articles on an end-to-end hyperspectral simulator, I found the manuscript very hard to read and follow. The idea of extending the simulator to include also a supervised thematic classification is interesting, but introduces an further complexity. Although the dependence of the outcome of classification on the system and acquisition parameters is mostly known, the added value of the extended system may be of interest, though for few readers.

For a deeply revised version of the manuscript, I recommend

1) insert a table of symbols. 

Response: Done. We have added a table of symbols after introduction.

2) insert a table of acronyms.

Response: Done. We have added a table of acronyms after introduction.

3) use suitable typesetting to indicate scalars, vectors, matrices, dependence on space  and on wavelength; in the present manuscript, it is not allowed to know is the wavelength lambda is a continuous or a discrete variable. I checked the notation of some articles that have been cited and it is much better.

Response: Done. Thanks for your advice. We have revised all the problems such as scalars, vectors that you mentioned in the whole article. And the wavelength lambda in our article is discrete. We have added a nomenclature chart that you suggested. Such as    the mean reflectance and covariance can be calculated be calculated with its reflectance of the training samples based on the ground class model.is the path radiation

4) Fig. 1 should possibly be moved at the end of Sect. 2 (not mandatory if the symbol table is clear and complete).

Response: Done. We have moved Fig.1 to the end of Section 2 as you suggested. It is much clearer than before and we also checked the symbols one by one to make sure all of them corrected.

5) use compact symbol names, otherwise a formula will be unreadable. I wish to remark that most of equations are hardly readable also for an expert, due to poor notation.

Response: Done. Thanks for your advice. We have revised the formulas in order to make all of them readable. And the chart at the beginning includes most of symbols we used in these formulas.

6) The reference list should be extended to comprise other upcoming hyperspectral missions (not only EnMap) and possibly the simulators of their instruments.

Response: Done. We have added some articles that related to some upcoming hyperspectral mission and the simulators of their instruments such as in Line 36-39 and Line 465-471.

Line 36-39 And there are some other upcoming hyperspectral mission and the simulators. Before the satellite launching, it is necessary to simulate and evaluate the process by some end-to-end simulators like EnMap [3], EeteS [4] and FLEX [5].

Line 465-471

   3              Guanter L , Kaufmann H , Segl K , et al. The EnMAP Spaceborne Imaging Spectroscopy Mission for Earth Observation[J]. Remote Sensing, 2015, 7(7):8830.

   4              Hofer S . EeteS - The EnMAP End-to-End Simulation Tool[J]. IEEE Journal of Selected Topics in Applied Earth Observations & Remote Sensing, 2012, 5(2):522-530.

   5              Rivera J P , Sabater N , Tenjo C , et al. Synthetic scene simulator for hyperspectral spaceborne passive optical sensors. Application to ESA's FLEX/sentinel-3 tandem mission[C]// Workshop on Hyperspectral Image & Signal Processing: Evolution in Remote Sensing. IEEE, 2017.

There are some graphics and tables can not be displayed clearly in this area. So we upload a word in order to show you the details that we revised.

Reviewer 2 Report

The manuscript introduces a combined model to simulate scenes, physical behaviour of sensors and image processing with the aim of assessing capability of hyperspectral imagery from satellite with respect to the simulation parameters. To this purpose a statistical indicator is resorted based on mean and covariance. 

Whereas the model is general enough to handle scene parameters (e.g., SZA, climate model), sensor parameters (e.g., number of spectral bands, SNR) and images processing methods, authors concentrate their analysis as an example on the Hymap sensor. In addition they choose as a reference a specific application: detection of minerals from images.

Authors also point out the limits of the model, leaving more complete settings to future generalizations (probably an important role should be played by the covariance among bands, as pointed in Sec. 4.2C)

Remarks:

Authors' approach is to vary only one parameter at a time with respect to a reference status. They mention that assessment with two or more parameters simultaneously varying is left to future investigations. However a particular role is played by the statistical (or machine learning) tool used for final processment of the analysis, indeed the only step of a real image analysis. First of all authors do not mention anywhere what method was used for the simulation, the reader can only guess from Section 4.2C that a fixed number of features (NSF) is detected and Principal Component Analysis is resorted. Authors should better describe the methodological part of the image processing.

Related to my previous issue, it is not clear to me how authors handle types of minerals inside an image: it seems to me that they perform an unsupervised classification (that is without labelled priori information). In addition they do not attempt to actually recognize the exact type of minerals but only to discriminate between the classes, how well they separate between each other, independently of the goodness of the separation in terms of recognized mineral. Is this the case? Otherwise an indicator of accuracy of recognized mineral could be more appropriate, provided that a gold standard ("truth") is defined in some way. Authors should discuss on this. Moreover maybe a further scheme with the flow of the specific methodology for the image processig could help to understand better.

Even though surface classification is an important field in remote sensing, however there exist several different applications where totally different statistical or machine learning tools are used (e.g., cloud detection, estimate of atmospheric concentration of consituents), For these some of the conclusions could be even qualitatively different. Authors should point this limit.

In my opinion this kind of study can be useful to assess the relative performance in solving a particular application with respect to a reference state (as in the manuscript). In fact there is a wide use of simulated images and indeed in the manuscript the only insertion of real data is the reference image (as reflectance) from which all other ones are generated. It is more challenging to estimate how quantitatively the estimate obtained is close to the actual scenes. This could be made for different climatic or geometric conditions (e.g., solar zenith angle) with appropriate images. Whereas this full analysis cannot be the subject of the present manuscript, however authors are requested to comment.

Some misprints:

l. 108: influences instead of influence

l. 372: Principal Component instead of Principle Component

Author Response

A combined quantitative evaluation model for the capability of hyperspectral imagery

Manuscript ID: sensors-411757

Dear Professor,

Many thanks for the constructive and encouraging comments on our manuscript from three reviewers. In particular, we welcome and note your comment Whereas the model is general enough to handle scene parameters (e.g., SZA, climate model), sensor parameters (e.g., number of spectral bands, SNR) and images processing methods, authors concentrate their analysis as an example on the Hymap sensor. We enclose a carefully revised manuscript according to the comments and suggestions made. We provide an item-by-item response to all comments. The responses are start with “Response” which is included in “bold”. We have also made some non-requested minor typographical and readability edits.

We hope that these clarifications and revisions will now enable the paper to be accepted for publication in Sensors, and look forward to hearing from you soon.

Yours sincerely,

Na Li (on behalf of all authors)

Response:

Authors' approach is to vary only one parameter at a time with respect to a reference status. They mention that assessment with two or more parameters simultaneously varying is left to future investigations. However a particular role is played by the statistical (or machine learning) tool used for final processment of the analysis, indeed the only step of a real image analysis. First of all authors do not mention anywhere what method was used for the simulation, the reader can only guess from Section 4.2C that a fixed number of features (NSF) is detected and Principal Component Analysis is resorted. Authors should better describe the methodological part of the image processing.

Related to my previous issue, it is not clear to me how authors handle types of minerals inside an image: it seems to me that they perform an unsupervised classification (that is without labelled priori information). In addition they do not attempt to actually recognize the exact type of minerals but only to discriminate between the classes, how well they separate between each other, independently of the goodness of the separation in terms of recognized mineral. Is this the case? Otherwise an indicator of accuracy of recognized mineral could be more appropriate, provided that a gold standard ("truth") is defined in some way. Authors should discuss on this. Moreover maybe a further scheme with the flow of the specific methodology for the image processig could help to understand better.

Even though surface classification is an important field in remote sensing, however there exist several different applications where totally different statistical or machine learning tools are used (e.g., cloud detection, estimate of atmospheric concentration of constituents), For these some of the conclusions could be even qualitatively different. Authors should point this limit.

In my opinion this kind of study can be useful to assess the relative performance in solving a particular application with respect to a reference state (as in the manuscript). In fact there is a wide use of simulated images and indeed in the manuscript the only insertion of real data is the reference image (as reflectance) from which all other ones are generated. It is more challenging to estimate how quantitatively the estimate obtained is close to the actual scenes. This could be made for different climatic or geometric conditions (e.g., solar zenith angle) with appropriate images. Whereas this full analysis cannot be the subject of the present manuscript, however authors are requested to comment.

1) Authors do not mention anywhere what method was used for the simulation, the reader can only guess from Section 4.2C that a fixed number of features (NSF) is detected and Principal Component Analysis is resorted. Authors should better describe the methodological part of the image processing.

Response: Done. Thanks for your advice. While processing there are two influence factors, mineral identification and feature extraction. The mineral identification didn’t affect the separability. We only take the influences of feature extraction into consideration. So that we have put forward the value of NSF in order to analyze quantitatively. The NSF values can be detected after feature extraction.

Line 404-409 A high correlation was observed among the adjacent bands in the hyperspectral narrow-band spectral data. In the case of the number of training samples being determined, the application capability will decrease as the bands of hyperspectral data increase. Therefore, hyperspectral data will be pre-processed with the dimension reduction model. However, the NSF must be calculated with different feature selection and extraction methods. The NSF values can be detected after feature extraction.

2) It is not clear to me how authors handle types of minerals inside an image: it seems to me that they perform an unsupervised classification (that is without labelled priori information). In addition they do not attempt to actually recognize the exact type of minerals but only to discriminate between the classes, how well they separate between each other, independently of the goodness of the separation in terms of recognized mineral. 

Response: Done. Thanks for your advice. First of all, the types of minerals that we chose were on the basis of ground survey of research area that we made before. At the meantime, the analysis of the datasets while processing in our article mainly discussed the influences of signal statistics (such as mean and covariance) which were affected by different parameters and algorithms. And because of the classification algorithm did not change the statistical parameters of the signal no matter whether it was supervised or unsupervised. Therefore, the evaluation index in our research is normalized mean distance (D) that we mainly discussed. So we didn’t consider classification methods or mineral identification in our proposed model.

3) However there exist several different applications where totally different statistical or machine learning tools are used (e.g., cloud detection, estimate of atmospheric concentration of constituents), For these some of the conclusions could be even qualitatively different. Authors should point this limit.

4) It is more challenging to estimate how quantitatively the estimate obtained is close to the actual scenes. This could be made for different climatic or geometric conditions (e.g., solar zenith angle) with appropriate images. Whereas this full analysis cannot be the subject of the present manuscript, however authors are requested to comment.

Response: Done. Thanks for your advice. By using different parameters we mentioned in the article, we can obtain different means and covariance. So that, because of different μ and Σ, the result of formula (15) is different. For example according to different parameters such as the image values simulated by different SZA, the means and covariance between classes obtained are different. So as to obtain different evaluation indexes to judge the impact of different SZA. And the atmospheric model and aero model are determined by the study area. The conditions are certain. As we discussed the application capability is a general definition. We proposed seven parameters are partial parameters while imaging. But these parameters are representative. While using these parameters in multi-spectral or RGB images to judge the application capability, some parameters should be different. What we considered is under ideal imaging conditions. But some non-ideal parameters might also influence the application capability such as distortion. That might be take part in our future research.

Some misprints:

l. 108: influences instead of influence

Response: Done. We have revised the typos. And we have also checked whole article to make sure there is no similar mistakes any more.

Line 212-213 The imaging condition model affects the at-sensor radiance and influences the application capability.

l. 372: Principal Component instead of Principle Component instruments.

Response: Done. We have revised the typos. And we have also checked whole article to make sure there is no similar mistakes any more.

Line 402-403 As the principal component analysis method was used to extract spectral features in our work

Reviewer 3 Report

The authors take on a very ambitious task: determining the  application capability of hyperspectral data is very hard, and combining this with a simulation of the imaging process with a complex model including many parameters.  They did a good effort, but inevitably could not cover all this in sufficient detail. A whole series of results are presented in which the variation of a single parameter results in changes in performance, mostly in line with common sense expectations.

The more interting questions arenot touched upon: What happens when multiple variables come into play, are the results correlated ?  Also: the effects shwon for this specific setup, can we still learn somethign from them in a more general sense ?  

Soem detailed comments:

190  The quantisation noise is correlated with the other random noise, so your model is not correct.  How did you determine  reasonable values for the covariance of the quantisation noise ?

247 This is very vauge; Do you apply the model as presented ? If so which parameters are used?

261 what are 'DISORT streams' ?

295: MTf at Nyquist, as you point out, is in fact a measure of aliasing.  The MTF at lower frequencies is what counts.  Which MTF model did you use to set a value for all frequencies ?

fig 3, 4 and 5:  a star diagram is not a good way of presenting the relation between SZA and evaluation index. Simple X_Y charts are much clearer.

Author Response

A combined quantitative evaluation model for the capability of hyperspectral imagery

Manuscript ID: sensors-411757

Dear Professor,

Many thanks for the constructive and encouraging comments on our manuscript from three reviewers. In particular, we welcome and note your comment They did a good effort, but inevitably could not cover all this in sufficient detail. We enclose a carefully revised manuscript according to the comments and suggestions made. We provide an item-by-item response to all comments. The responses are start with “Response” which is included in “bold”. We have also made some non-requested minor typographical and readability edits.

We hope that these clarifications and revisions will now enable the paper to be accepted for publication in Sensors, and look forward to hearing from you soon.

Yours sincerely,

Na Li (on behalf of all authors)

Response:

The authors take on a very ambitious task: determining the  application capability of hyperspectral data is very hard, and combining this with a simulation of the imaging process with a complex model including many parameters.  They did a good effort, but inevitably could not cover all this in sufficient detail. A whole series of results are presented in which the variation of a single parameter results in changes in performance, mostly in line with common sense expectations.

The more interesting questions are not touched upon: What happens when multiple variables come into play, are the results correlated ?  Also: the effects shwon for this specific setup, can we still learn something from them in a more general sense ?  

Response: Done. Thanks for your advice. There are the correlations or coupling relationships between imaging parameters for hyperspectral imaging system. In our proposed model, the multi-parameters of imaging system could be considered. If we have the quantitative expression equation or analytic description for the relationships of those parameters, we can improve our model to consider the joint effects. Therefore, our recent research or our proposed model has some reference or some results correlated and we can still learn something from the effects in a general sense (such as the sensitivity of different imaging parameters for the application capabilities). We have some initial research about the coupling or constraint relationship shown as reference [1]:

[1] Na Li; Xinchen Huang; Huijie Zhao, Multi-parameters Optimization for Mineral Mapping using Hyperspectral Imagery, IEEE Journal of Selected Topics in Applied Earth Observations and Remote Sensing, 2018, 11(4): 1348-1357.

In our future work, we will build the joint-analysis model based on our recent proposed model and the constraint relationship between imaging parameters.

Some detailed comments:

1) 190 The quantization noise is correlated with the other random noise, so your model is not correct.  How did you determine  reasonable values for the covariance of the quantisation noise ? 

Response: Done. Thanks for your advice. The random noise includes quantization noise. In our research, the different random noises (such as photon noise, read-out noise and quantization noise) were analyzed separately, however, they are unified into random noises in the calculation process and calculated based on the sensors parameters. The details revised in our manuscript are in Line191-195

Line 191-195 The covariance of random noises can be calculated by using their statistical distribution characteristics, and they can be added directly. is the signal covariance vector after noise model and it can be calculated as follows:

where, , , and  are the covariance matrices of photon noise, read-out noise, and quantization error, respectively, and  is the covariance matrix.

2)247 This is very vague; Do you apply the model as presented ? If so which parameters are used?

Response: Done. Thanks for your advice. We have revised these parts and add some details about the simulation method for the generation of simulated data. The details revised in our manuscript are in Line 259-276.

Line 259-276 In order to analyze the different influence factors of application capability, which are caused by the imaging system parameters, a total of 50 images were simulated using a satellite-borne hyperspectral image simulation method. A Hymap reflectance image was used as the input scene. The several processing steps were used to generate the simulated images, including the creation of atmospheric effects using an improved radiation transfer model that considers topographic effects[17], the satellite-borne imaging spectrometric model which incorporates the spectral, radiative, and spatial features of the hyperspectral imager [9], and the simulation of a set of data with different imaging system parameters (SNR, SR, GSD, SRF, and MTF) under the same atmospheric and imaging conditions and with different SZA under the same imaging system parameters were simulated in the simulation experiment. And then the mean and covariance with different parameters were calculated to obtain the different the evaluation index of application capability based on our proposed combined evaluation model.

3)261 what are 'DISORT streams' ?

Response: Done. ‘DISORT streams’ means Discrete Ordinates Radiative Transfer. It is one of the atmospheric conditions that we set when simulating. The ‘parameter value 8’ means the discrete algorithm estimating eight scattering directions.

4) 295: MTf at Nyquist, as you point out, is in fact a measure of aliasing.  The MTF at lower frequencies is what counts.  Which MTF model did you use to set a value for all frequencies?

Response: Done. In the section 2.3.1 spatial model and analytic description, the MTF was introduced to measure the image degradation (especially aliasing), and the MTF can be calculated by the Fourier transform of point spread function of spatial response. The Gaussian model was introduced to describe the distribution of the point spread function. Therefore, the Gaussian model was used to describe the MTF. The added descriptions were shown in Line 165-167.

Line 165-167 The Modulated Transfer Function (MTF) could be calculated by the Fourier transform of point spread function. Therefore, the GSD and MTF were introduced to analyze the effects of the spatial model for application capability.

5) fig 3, 4 and 5:  a star diagram is not a good way of presenting the relation between SZA and evaluation index. Simple X_Y charts are much clearer

Response: Done. We had added the X_Y charts to describe the relation between different parameters and evaluation index. And they were shown in Figure3(b), 4(b)-4(f), and 5(b).

The graphics can not be displayed in this area. So we upload a WORD in order to show the details that we revised.
